# Precipitation Dominates the Allocation Strategy of Above- and Belowground Biomass in Plants on Macro Scales

**DOI:** 10.3390/plants12152843

**Published:** 2023-08-01

**Authors:** Xianxian Wang, Xiaohong Chen, Jiali Xu, Yuhui Ji, Xiaoxuan Du, Jie Gao

**Affiliations:** 1College of Life Sciences, Xinjiang Normal University, Urumqi 830054, China; Yechel720@163.com (X.W.); cheng1352023@163.com (X.C.); m18099534368@163.com (J.X.); jyh1757838695@163.com (Y.J.); 2Coastal Agriculture Research Institute, Kyungpook National University, Daegu 41566, Republic of Korea; 3Key Laboratory of Earth Surface Processes of Ministry of Education, College of Urban and Environmental Sciences, Peking University, Beijing 100871, China

**Keywords:** global change, biomass allocation, functional traits, soil nutrients, type of plants

## Abstract

The allocation of biomass reflects a plant’s resource utilization strategy and is significantly influenced by climatic factors. However, it remains unclear how climate factors affect the aboveground and belowground biomass allocation patterns on macro scales. To address this, a study was conducted using aboveground and belowground biomass data for 486 species across 294 sites in China, investigating the effects of climate change on biomass allocation patterns. The results show that the proportion of belowground biomass in the total biomass (BGBP) or root-to-shoot ratio (R/S) in the northwest region of China is significantly higher than that in the southeast region. Significant differences (*p* < 0.05) were found in BGBP or R/S among different types of plants (trees, shrubs, and herbs plants), with values for herb plants being significantly higher than shrubs and tree species. On macro scales, precipitation and soil nutrient factors (i.e., soil nitrogen and phosphorus content) are positively correlated with BGBP or R/S, while temperature and functional traits are negatively correlated. Climate factors contribute more to driving plant biomass allocation strategies than soil and functional trait factors. Climate factors determine BGBP by changing other functional traits of plants. However, climate factors influence R/S mainly by affecting the availability of soil nutrients. The results quantify the productivity and carbon sequestration capacity of terrestrial ecosystems and provide important theoretical guidance for the management of forests, shrubs, and herbaceous plants.

## 1. Introduction

Plant biomass is an important indicator for quantifying the productivity of terrestrial ecosystems [1,2,3]. The ratio of belowground biomass (BGB) to aboveground biomass (AGB), known as the root/shoot ratio (R/S), and the proportion of belowground biomass in the total biomass (BGBP) are both used to reflect the optimal allocation of resources by plants [4]. Under resource-rich conditions, plants allocate more biomass to aboveground photosynthetic organs, increasing their competitiveness for light to acquire more resources [5]. Correspondingly, under harsh environmental conditions, plants enhance the utilization efficiency of their root systems for water and soil nutrients by increasing their BGB [6]. Although preliminary research results on AGB-BGB allocation patterns have been obtained in some forest and grassland ecosystems [7], the allocation patterns of aboveground and belowground biomass and their controlling factors on a larger spatial scale remain unclear due to limitations in spatial and temporal scales of specific regions. [8,9]. Therefore, is of great importance to explore the distribution pattern of plant biomass, an important ecological parameter and the third-largest carbon pool in terrestrial ecosystems [10], in promoting plant resource acquisition and growth.

Climate change significantly affects the allocation patterns of aboveground and belowground plant biomass as well as the resource utilization strategies of plants [11,12]. With global climate change, plants adjust their aboveground and belowground biomass allocation strategies to maximize the use of limited resources, improving their survival chances and responding to climate change [13]. As temperature increases, soil temperature also rises, indirectly increasing root turnover and decomposition rates, promoting plants to allocate more biomass aboveground for rapid growth [5]. Plants in high-latitude regions, which experience long-term low-temperature environments, increase their BGB to ensure they obtain enough soil nutrients to improve their chances of survival [14]. This is mainly because plants increase their R/S to compensate for the low availability of soil nutrients in cold environments [15]. Studies also show that the impact of temperature on biomass depends on the availability of soil moisture [16]. To acquire more water and nutrients from the soil, plants invest more dry matter in strengthening root development [17]. Particularly in arid regions, plants face the dual constraints of high temperature and water shortage, leading to an increased allocation of BGB [2,3,18].

The biomass allocation of different life types of plants is also closely related to changes in soil nutrients [19]. Plants absorb water and mineral elements from the soil, which provides the nutrients needed for plant growth, development, and reproduction [20]. The allocation strategy of aboveground and belowground plant biomass is constrained by soil nitrogen (N) and phosphorus (P) content [21]. Optimal allocation theory suggests that in nutrient-poor soil environments, plants allocate more biomass to the roots than the aboveground parts [22]. As soil fertility increases, the proportion of biomass allocated to the aboveground parts of plants will significantly increase [23]. Previous research has found that adding N and P to nutrient-poor soil environments weakens the competition between root systems and soil microorganisms due to higher soil N content [24]. The N addition is beneficial for increasing soil nutrients, prompting plants to allocate more biomass aboveground to capture more light energy [25]. When belowground resources are insufficient, plants invest more biomass into their root systems to acquire more soil nutrients and water, often resulting in higher BGB or R/S [26].

In recent years, plant functional traits have been widely used to predict changes in AGB-BGB allocation at macro scales [27,28]. Plant functional traits are adaptive physiological and morphological characteristics formed by plants in response to environmental changes [29]. Ali et al. [30] found in their study of tropical forests that plant AGB is positively correlated with specific leaf area (SLA) and negatively correlated with leaf N content (LN). To adapt to ecological environment changes, plants exhibit traits such as higher LN, larger leaf area, and lower AGB [31]. Similarly, in harsh, arid environments, plants increase their investment in BGB, accumulate large amounts of leaf dry matter content (LDMC), and enhance their stress resistance [32]. Plants with higher LDMC and lower SLA adopt a more “conservative” resource acquisition strategy, allocating more biomass below ground [33]. Changes in plant organ morphological traits are ultimately manifested as changes in AGB-BGB allocation [33]. Plants increase their biomass in leaves by enhancing their photosynthetic capacity to acquire more organic matter [34].

Different plant life types have distinct life history strategies [35]. Previous research has shown that the allocation of aboveground and belowground biomass in plants is also influenced by growth form differences [35]. Shrub species, with their higher regenerative capacity, have a greater ability to adapt their aboveground and belowground biomass allocation patterns to environmental changes compared to forests [36]. Herbaceous plants allocate 80% of their biomass belowground, which is more than what trees allocate [37]. Studies have found that in warm and humid environments, grassland biomass is mainly limited by soil nutrients, while under arid conditions, precipitation limits grassland net primary productivity (NPP) [38]. NPP refers to the total accumulation of organic matter and dry matter in plants per unit time and unit area [39]. Plant biomass and NPP are two key factors determining terrestrial ecosystem carbon cycling [40]. In an experiment on grassland NPP, Tateno et al. used the peak AGB as a representation of aboveground NPP in plants [41]. Research has shown that the spatial allocation of forest NPP, rather than temperature and precipitation, is the main factor controlling forest biomass in the southwestern region of China [42]. Ecosystems with higher NPP promote plants to allocate more biomass to the aboveground parts, which is beneficial for capturing more sunlight [43]. An increase in plant biomass and the number of dead branches and leaves leads to higher NPP in forests [41]. Therefore, exploring the biomass allocation strategies of different life types and their relationships with NPP is crucial for understanding carbon storage and carbon cycling in ecosystems [42].

In this study, based on AGB-BGB data from 486 species at 294 sites across the country, we explored the patterns of biomass allocation in different life types of plants and their influencing factors. To answer these questions, we proposed the following hypotheses. (1) At the macro scale, there are significant differences in BGBP or R/S among different life types of plants. (2) Environmental factors (climate factors, soil nutrient factors) have a greater relative contribution to driving plant biomass allocation patterns than plant functional traits. (3) Climate factors indirectly affect plant biomass allocation patterns by acting on soil nutrient factors and plant functional traits.

## 2. Results

The aboveground and belowground biomass allocation patterns of plants with different life types show significant geographical differences, and their trends change consistently with latitude and longitude (Figure 1). The BGBP and R/S of plants with different life types gradually decrease from northwest to south. Lower R/S is allocated in the southwest and northeast regions of China (Figure 1b). The differences in BGBP or R/S of trees, shrubs, and herbs were significant (*p* < 0.001, Figure 2). The BGBP and R/S of herb plants are significantly higher than that of shrubs and trees.

We found significant correlations among potential influencing factors of BGBP and R/S (climatic factors, soil factors, and functional trait factors). Overall, the effects of climatic factors on BGBP and R/S are greater than those of soil factors and functional trait factors (Figure 3 and Appendix A). Random forest analysis showed that life type and NPP were the main predictors of BGBP in plants, while MAP was the main predictor of R/S (Figure 4). Precipitation and soil nutrients (soil N content and soil P content) have positive effects on BGBP or R/S. However, MAT, NPP, and functional traits were negatively correlated with BGBP or R/S (Figure 5).

SEM shows that climate factors play an important role in regulating BGBP or R/S (Figure 6). Climate change indirectly regulates BGBP by changing plant functional traits. (Figure 6a). We also found that climatic factors determine R/S by altering the effectiveness of soil nutrients (Figure 6b). Although BGBP and R/S show similar trends with changes in individual influencing factors, there are significant differences in the pathways through which climatic factors affect them.

## 3. Discussion

The biomass allocation pattern of plants in China shows significant geographical differences, which may be closely related to climatic factors. Climatic factors indirectly affect plant growth by directly influencing the allocation of aboveground and belowground biomass [44]. Previous studies have found that lower water use efficiency in plants can lead to higher R/S [45]. When water resources are scarce, plants allocate more biomass to the belowground part to maximize belowground resource acquisition [25]. A drier climate optimizes root investment and maintains a functional balance between above- and belowground resources [46] Similarly, the growth rate of plants is strongly influenced by low temperature, and as the temperature increases, plants allocate more biomass to branches and leaves for rapid growth [15]. China’s terrain is higher in the west and lower in the east, and precipitation gradually increases from west to east [32,47]. Therefore, lower R/S appears in the southwest and northeast regions of China. Due to the heterogeneity of habitats in different regions, there are significant differences in microenvironments (Appendix A). Previous studies have found that habitat heterogeneity significantly affects the biomass allocation pattern of plants [39,48].

The biomass allocation pattern of plants is closely related to their life type [49]. Plant morphology affects the allocation pattern of aboveground and belowground biomass [35], and the differences in BGBP or R/S among trees, shrubs, and herb plants are highly significant (*p* < 0.001). Previous studies have found that compared to trees, shrubs with higher R/S are more adaptable to adverse environments [19]. Shrub plants allocate more biomass to the belowground part to obtain more resources for growth due to their strong regenerative capacity and developed root system [36]. In semi-arid grasslands, where biomass is limited by water, herbs plants allocate three-quarters of their biomass to belowground parts to absorb limited water resources [37]. Compared to shrubs and herb plants, trees allocate much less biomass to the belowground part [26].

The difference in the aboveground and belowground biomass allocation of plants is closely related to climate and soil nutrients [21]. Climatic factors affect plant growth by changing their resource acquisition strategy [4]. Temperature not only directly affects the activity of enzymes related to plant photosynthesis and respiration, but also affects biomass production [39]. An increase in environmental temperature enhances the effectiveness of water and soil nutrients, leading to reduced investment in belowground biomass by plants and rapid growth and development [50]. Climate warming has been shown to affect biomass production in terrestrial ecosystems [51]. The availability of soil moisture also depends on MAP, and as precipitation decreases, plants allocate more biomass to roots to increase water uptake [10]. Precipitation can increase soil moisture, and the increase in soil moisture promotes root growth and releases root exudates, indirectly enhancing organic matter degradation and soil microbial activity [52]. Higher precipitation also increases soil N mineralization and plant N use efficiency, increasing plant AGB [14]. However, biomass allocation strategies are also significantly affected by soil nutrients [16]. Habitat with limited soil N and P can lead to an increase in plant BGB [38]. When plants are deficient in P and N, they accumulate large amounts of carbohydrates in leaves and allocate more carbon to the roots, leading to an increase in R/S [53]. An increase in soil N and P availability can offset the negative effects of water shortage and drought on BGB in high-temperature environments [27]. China’s vegetation types are diverse, from tropical rainforests to desert steppes in the northwest, and our results span a wide range of environmental gradients and geographic scales (Appendix A), indicating that climate factors and soil factors are the main drivers of plant biomass allocation [12,22].

Numerous studies have shown that plant life type, functional traits, climate, soil nutrients, and local NPP collectively affect the allocation of plant aboveground and belowground biomass [11,23,45]. To adapt to changing environmental conditions, plants adjust their morphological traits to obtain more resources, ultimately resulting in biomass accumulation [20,34]. The differences in aboveground and belowground biomass allocation strategies of plants reflect differences in species’ resource acquisition strategies [54]. The influence of climate on plant aboveground and belowground biomass allocation is mainly manifested through the corresponding evolution of plant functional traits [44]. In cold and arid ecosystems, plants enhance their photosynthetic capacity by changing leaf traits to increase LN, while allocating more biomass to root growth to improve water uptake efficiency [13,24]. In addition, plants also reduce SLA and increase LDMC to resist drought and prolong leaf life, while increasing BGB to absorb nutrients and extend the lifespan of the entire plant [7]. Chai et al. [49] found that plant traits explained the variation in AGB, while climate played an important role in the variation in R/S in different ecological regions. Climate factors mainly determine the allocation of aboveground and belowground biomass in terrestrial ecosystems in China by altering plant functional traits. In the arid regions of northwest China, plants’ conservative resource acquisition strategies often accompany higher BGB or R/S, while the aboveground and belowground biomass allocation patterns in areas with abundant water and heat are opposite [1].

## 4. Conclusions

Based on 1337 biomass data from 294 sites in the database, we verified the driving factors of climate, soil nutrients, functional traits, and NPP on the biomass allocation strategy of different plant life types. The results showed that BGBP or R/S in northwest China was significantly higher than that in the southeast region. The BGBP or R/S of herbs plants was significantly higher than that of shrubs and trees. Climate factors were the main driving factors shaping the above- and belowground biomass allocation patterns of different plant life types. Climate factors determined the BGBP of plants by changing their functional traits, while the variation in R/S was determined by climate factors changing the soil nutrient factors. Therefore, paying attention to the effects of climate change drivers on aboveground biomass allocation in China is crucial for predicting the response of terrestrial ecosystems to global change. At the same time, in the context of global environmental change, this study, on the one hand, provides important theoretical guidance for understanding the above- and belowground biomass allocation strategies of different plant life types, and on the other hand quantifies the productivity and carbon sequestration capacity of terrestrial ecosystems.

## 5. Methods

### 5.1. Data Collection

We searched the China National Knowledge Network and Web of Science databases using “biomass” and “biomass allocation” as keywords to collect relevant literature on the aboveground and belowground biomass allocation of trees, shrubs, and herbs in China between 2000 and 2022. According to the text, table, and appendix data provided in each article, the units of AGB and BGB data of each species were uniformly converted into kg m^−2^, and then included in the dataset. These articles were screened strictly according to the following four criteria: (a) all the plants were grown in the wild and under natural conditions without human interference; (b) AGB and BGB were measured simultaneously, and both are dry weight values obtained after oven drying; (c) when AGB is in the strong growing season, BGB used the excavation method for destructive cutting, excluding the sampling depth of trees and shrubs less than 1 m and all the data estimated by the model; (d) all values can be extracted directly from tables or text or appendices. Finally, 324 articles were selected (Appendix A).

The database includes 294 locations, 174 tree species, 150 shrub species, and 162 herb species. Considering the spatial heterogeneity of different literature data, different sampling sites, and individual size differences between plants, we carried out a unified conversion process for the collected AGB and BGB data, and used the ratio of underground biomass of each plant to the total biomass, BGBP (ratio of underground biomass to the total biomass). The percentage of biomass values corresponding to each plant is used to represent the biomass of different living plants, eliminating the individual size differences of different plants. The BGBP (%) and R/S are calculated as follows:(1)$$BGBP=\frac{BGB}{AGB+BGB}\times 100\%$$
(2)$$R/S=\frac{BGB}{AGB}$$

Our database information also includes geographical coordinates (longitude, latitude, and altitude), climatic factors (mean annual temperature (MAT), °C; mean annual precipitation (MAP), mm), soil nutrient factors (soil N and P content, g kg^−1^), NPP (kg cm^−2^ a^−1^), and functional trait data. Functional trait data include tree height (m), leaf length (mm), leaf width (mm), leaf N and P content (g kg^−1^), root N and P content (g kg^−1^), stem N and P content (g kg^−1^), and litter N and P content (g kg^−1^). The vast majority of variable data were extracted from the literature. For missing data, we extracted MAT and MAP based on the latitude and longitude of the study sites from the China Meteorological Administration Meteorological Data Center (http://data.cma.cn/site/index.html, accessed on 1 May 2023) and extracted topsoil N (http://www.csdn.store, accessed on 1 May 2023) and available P (https://www.osgeo.cn/data/wc137, accessed on 1 May 2023) data from 250 m resolution grids. The required NPP was extracted from the 500 m resolution NPP contribution data for China’s region from 2001 to 2022. These data comprise the longest NDVI data series and have been widely used for the estimation and investigation of large-scale vegetation dynamics, vegetation NPP, and biomass. The plant functional trait data come from the dataset published by Zhang et al. [55] (https://doi.org/10.5194/essd-13-5337-2021, accessed on 1 May 2023).

### 5.2. Data Analysis

We used the kernel density estimation method to draw the allocation pattern of plant BGBP (%) or R/S of trees, shrubs, and herbs in China, with a spatial resolution of 1 × 1 km [38].

We used a significance test with a 0.05 significance level to test whether there are significant differences in BGBP (%) or R/S for plants with different life types. We performed the significance test using the R software package agricolae (version 4.1.0, R Core Team, 2020).

We used one-way analysis of variance (ANOVA) to perform statistical analysis on BGBP (%) or R/S data. Prior to model development, principal component analysis (PCA) was performed for 11 selected functional trait factors most relevant to plant biomass allocation. The first principal component (PC1) was found to explain 43.7% of functional traits (Appendix A). Subsequent analysis used the Fisher least significant difference (LSD) test. *p* < 0.05 indicates a statistically significant difference. We visualized networks using Cytoscape [56]. The R package used was linkET.

To identify the main predictors of BGBP (%) or R/S in plants, a random forest analysis was performed for the relative importance of climate factors, soil nutrient factors, functional traits, and NPP. The random forest analysis was completed by the software packages randomForest and rfPermute in the R statistical computing environment.

To further explore the spatial variation mechanism of various factors on the influence of BGBP (%) or R/S, a piecewise structural equation modeling (piecewiseSEM) was used to assess the direct and indirect relationships between key ecosystem factors and BGBP (%) or R/S. We constructed a prior model based on the literature, considering geographical location (longitude, latitude), life type, climate, functional traits, soil nutrients, and NPP. At the initial stage of model construction, there was collinearity between geographical location information and life type information with BGBP (%) or R/S, which were subsequently removed from the final model. All observed variables were first divided into composite variables and then incorporated into SEMs. To confirm the robustness of the relationship between key ecological factors and BGBP (%) or R/S, we used piecewiseSEM to explain the random effects of the sampling points and provide the “marginal” and “conditional” contributions of environmental predictive factors [57]. These analyses were performed using the piecewiseSEM, nlme, and lme4 packages [57]. Fisher’s C-test was used to assess the goodness of fit of the modeling results. Based on the significance level (*p* < 0.05) and model goodness (0 ≤ Fisher’s C/df ≤ 2 and 0.05 < *p* ≤ 1.00), the model was modified stepwise.

## Figures and Tables

**Figure 1 plants-12-02843-f001:**
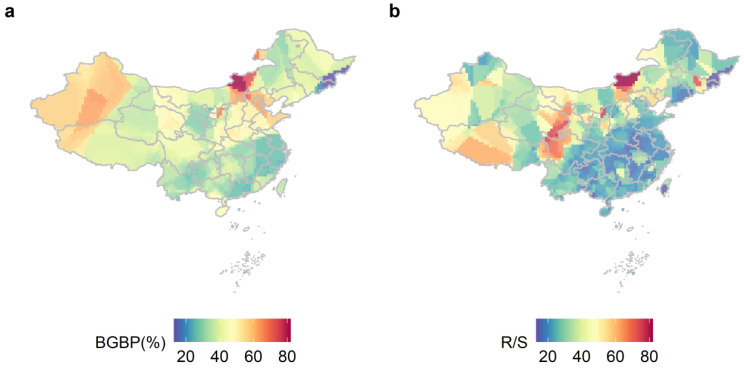
The allocation patterns of BGBP (%) (**a**) and R/S (**b**) for different life types in China were studied using kernel density estimation with a spatial resolution of 1 × 1 km. BGBP represents the ratio of belowground biomass to total biomass, while R/S represents the ratio of BGB to AGB.

**Figure 2 plants-12-02843-f002:**
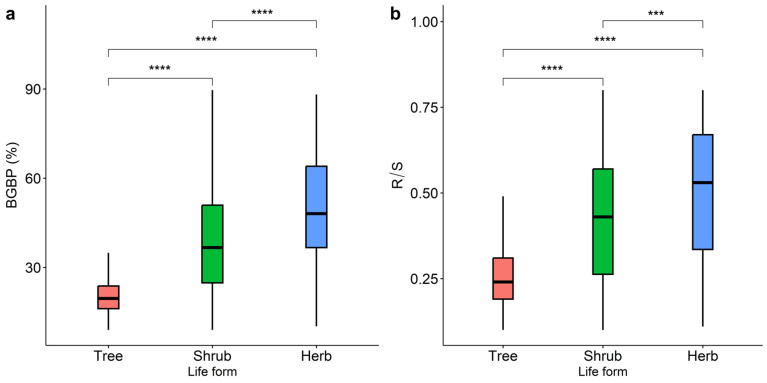
Comparison of the variation in BGBP (%) (**a**) and R/S (**b**) among trees, shrubs, and herbs. The symbols *** and **** indicate statistical significance at *p* < 0.001 and *p* < 0.0001, respectively.

**Figure 3 plants-12-02843-f003:**
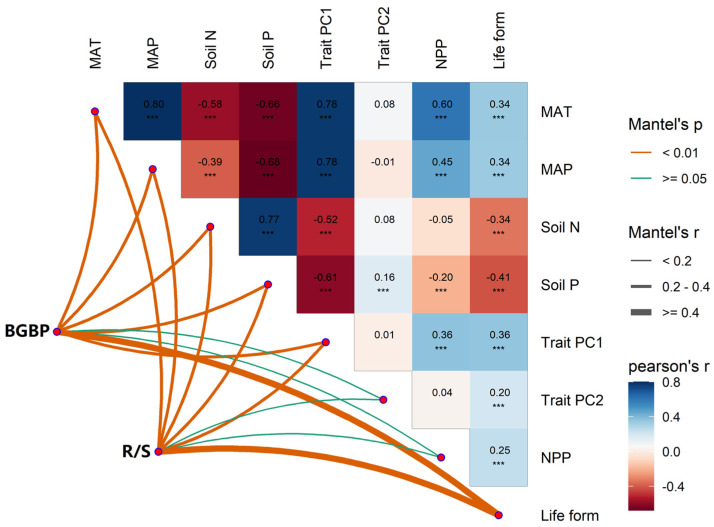
Multivariate correlation analysis was conducted to investigate the potential influencing factors of BGBP (%) and R/S for different life types of plants, including climate factors (MAT and MAP), soil nutrient factors (soil N content, soil P content), the first two major components of functional traits (Trait PC1 and Trait PC2), NPP, and life type. *** indicate statistical significance at *p* < 0.001.

**Figure 4 plants-12-02843-f004:**
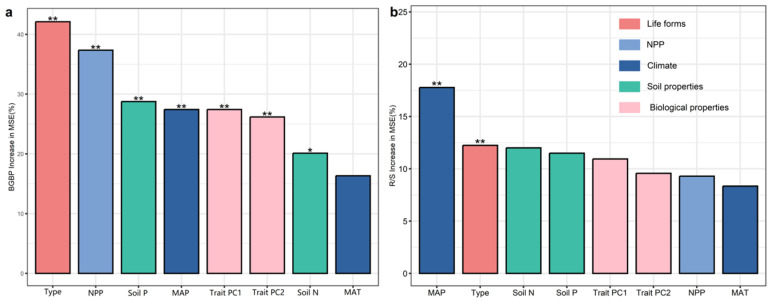
Main influencing factors for BGBP (%) (**a**) or R/S (**b**) of trees, shrubs, and herbs. The plot shows the significance of BGBP (%) or R/S for trees, shrubs, and herbs to the random forest mean predicted value (percentage increase in mean variance error (MSE)) for each impact factor. The colors of the bars represent different variable types. Significance level: ** *p* < 0.01; * *p* < 0.5. If it was not listed, it had no significance (*p* > 0.5).

**Figure 5 plants-12-02843-f005:**
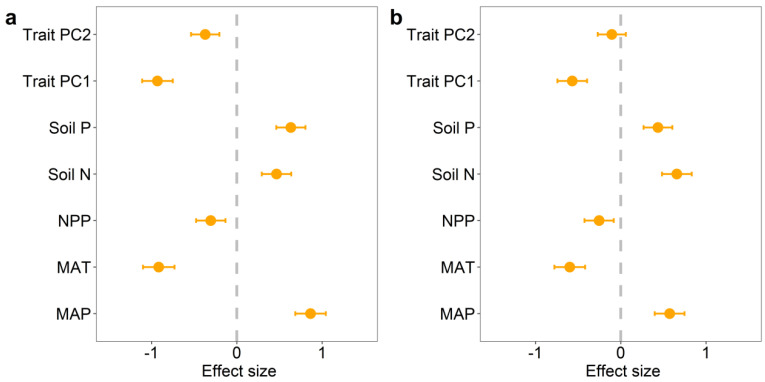
The impact of different treatments on plant BGBP (%) (**a**) and R/S (**b**). Effect size represents the positive or negative influence of the treatment on BGBP (%) and R/S. The treatments include the first two principal components of community functional traits (Trait PC1 and Trait PC2), soil nutrient factors (soil N content, soil P content), NPP, and climate factors (MAT and MAP).

**Figure 6 plants-12-02843-f006:**
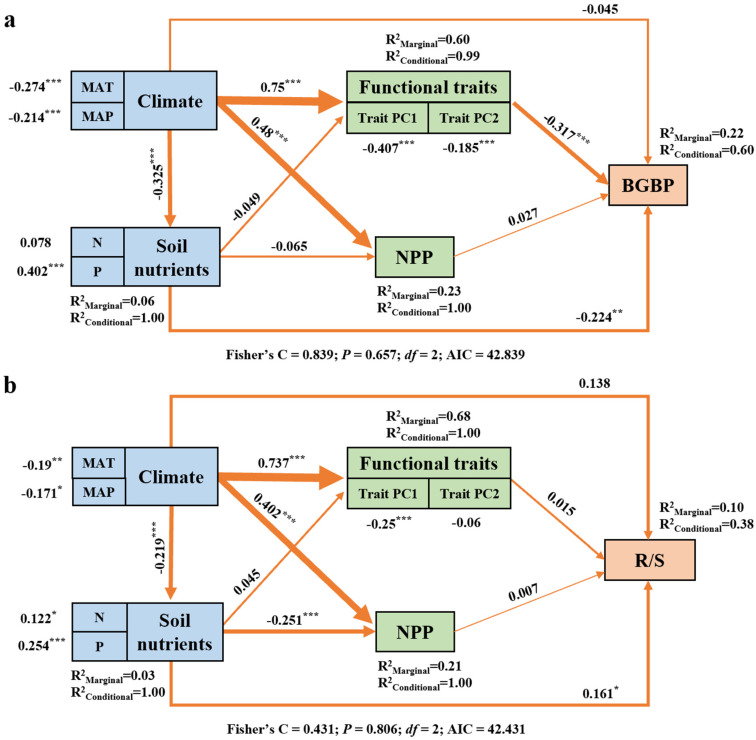
The association between climatic factors, soil nutrient factors, key functional traits, NPP, and BGBP (%) (**a**) and R/S (**b**) of different life types of plants. Path diagrams represent the standardized results of the final structural equation model (SEM) used to examine the relationships between the variables. The number adjacent to the arrow is the path coefficient, which is the direct normalized effect size of the relationship. The thickness of the arrow represents the strength of the relationship. Asterisks indicate significance (****p* < 0.001; ***p* < 0.01; **p* < 0.05). R^2^ indicates the goodness-of-fit of the generalized additive model (GAM). The best SEM with the lowest AIC was selected.

## Data Availability

Not applicable.

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
