# Peer review of "Precipitation Dominates the Allocation Strategy of Above- and Belowground Biomass in Plants on Macro Scales"

_plants, 2023, doi:10.3390/plants12152843_

Round 1

Reviewer 1 Report

This study investigated the effects of climate change on biomass allocation patterns in plants. Using data from 486 species across 294 sites in China, the study found that the proportion of belowground biomass in the total biomass was higher in the northwest region compared to the southeast region. The findings provide important insights into the impacts of global change on plant resource allocation strategies. In general, this paper is well-written and some minor issues should be addressed before publication.

1. Scientific questions are not well described in the introduction. Please highlight the necessity and importance of why this study was conducted.

2. Sample numbers are not provided in those Figures.

3. Supplementary information is not found in the system.

4. Authors should also add a Conclusion section.

5. Reference format is not consistent. Please check.

English should be improved by native speakers.

Reviewer 2 Report

This manuscript has a potential, but needs improvements. 

Generally, the English is good but same places are difficult to follow or are senseless altogether. 

The figures are not numbered and I had difficulty to identify which figure corresponded to which caption.

The authors confuse correlation with real, physical effects. Correlation might be an indication to such effects, but only a manipulative experiment can prove causation. Your study is purelly correlative. Please use "correlation", "association", "relationship" instead of "affects", "have an effect on", "impacts", "influences".

ABG, BGB andf R/S are traits too (compare with SLA, for example). So do not try to separate ABG, BGB or R/S from other traits. Just say "other functional traits such as...".

These and all other specific comments are in the attached file.

Needs careful editing by a native English-speaking colleague, or a professional editor who understands plant ecology. E.g., "herb plants" is a poor translation, sounds like we are talking about spice herbs such as coriander or parsley; must be "herbaceous plants"

Reviewer 3 Report

The manuscript is interesting and makes a significant contribution to the field of biomass allocation in plants. As a result, it has the potential to be published in the Plants journal. However, there are a few weaknesses and missing information that need to be addressed for the paper to reach its full potential. Therefore, I recommend a minor revision of the manuscript.

First of all, I have a complaint about the organization of the paper, particularly concerning the Figures and their related captions. The Figures lack order numbers, making it rather uncomfortable to couple a figure with its caption. The authors should consider improving this aspect in the future. Additionally, I did not find any Tables mentioned, which could be a part of the Appendix (as mentioned in line 236).

Specific comments on the individual sections:

Abstract:

I feel that the Abstract could benefit from a strong concluding statement that generalizes the scientific outputs of the work and/or provides an exact conclusion about the implementation of the results in any scientific field or practice.

Key words:

I recommend adding one more word - "type of plants." In fact, the term "life form" used in the main text and figures of the manuscript could be replaced with this word, which would be a much better expression.

Introduction:

The section is generally well-written, but I noticed a relatively small percentage of citations to sources other than Chinese works. It would be beneficial to include more diverse references.

Results:

The text is acceptable, although I have some suggestions regarding the Figures. The authors should always indicate the unit for each variable - for instance, the unit "%" is missing for BGSP in some cases. Additionally, the unit "g/kg" could be expressed in a more scientific way, such as "g kg-1" or "grams per kilogram."

As for Figure 2 and Figure S5, the letters indicating statistical significance should be larger. The current size is too small for comfortable reading.

Discussion:

I would appreciate a more exact explanation of the contribution of the results to science, particularly focusing on the specific implications for China in the context of ongoing climate change.

Conclusion:

Similar to my concern with the Discussion text, I would welcome more precise ideas about the practical contribution of the results and the necessity for further research in this field. The statements in the present form are somewhat vague.
